# UNPACKING LARGE LANGUAGE MODELS WITH CONCEPTUAL CONSISTENCY

## ABSTRACT

If a Large Language Model (LLM) answers "yes" to the question "Are mountains tall?" then does it know what a mountain is? Can you rely on it responding correctly or incorrectly to other questions about mountains? The success of Large Language Models (LLMs) indicates they are increasingly able to answer queries like these accurately, but that ability does not necessarily imply a general understanding of concepts relevant to the anchor query. We propose conceptual consistency to measure a LLM's understanding of relevant concepts. This novel metric measures how well a model can be characterized by finding out how consistent its responses to queries about conceptually relevant background knowledge are. To compute it we extract background knowledge by traversing paths between concepts in a knowledge base and then try to predict the model's response to the anchor query from the background knowledge. We investigate the performance of current LLMs in a commonsense reasoning setting using the CSQA dataset and the ConceptNet knowledge base. While conceptual consistency, like other metrics, does increase with the scale of the LLM used, we find that popular models do not necessarily have high conceptual consistency. Our analysis also shows significant variation in conceptual consistency across different kinds of relations, concepts, and prompts. This serves as a step toward building models that humans can apply a theory of mind to, and thus interact with intuitively.

## 1 INTRODUCTION

Large Language Models (LLMs) have had many exciting recent successes. These include high performance and even emergent capabilities using just zero or few-shot prompting (Brown et al., 2020; Wei et al., 2022a), but overall performance is still low compared to humans on a wide range of tasks for even the largest models (Srivastava et al., 2022), and our understanding of these models work is still limited. A popular explanation of low performance and inconsistencies is that LLMs are simply learning to mimic the data used to train them, and this basic pattern recognition limits generalizability, in the case of LLMs exposing the limits of any understanding (Zhang et al., 2022a; Bender & Koller, 2020). We would use a similar line of reasoning to guess how a LLM would answer the following question: "Can GPT-3 see?"

If it performed well on examples from the same distribution we would say it is likely to get it right or vice-versa if performed poorly on those examples. Though valid, this explanation is incomplete because it is completely agnostic to the specific content of the statement. We would apply the exact same reasoning and come to the same conclusion for similar statements about say blood banks or mock trials, as long as they were from the same distribution (in this example, the CSQA2 dataset (Talmor et al., 2021)). This is in contrast to our day to day life, where our Theory of Mind allows us to understand other agents (people) by attributing beliefs, intentions, and desires to them (Premack & Woodruff, 1978) in a way that allows us to usefully predict their behavior (Rabinowitz et al., 2018; Dennett, 1991). Beliefs are most relevant here, and should be conceptual in order to best support human understanding (Yeh et al., 2021). Ideally we would also be able to apply this kind of understanding to LLMs, predicting that the model is more likely correct about GPT-3's sight if it knows about GPT-3 generally than if it does not. This would be a conceptual model of the LLM that allows us to predict its behavior.

| | Irrelevant Background: Is the sky blue? | Relevant Background: Was GPT-3 built by OpenAI? | Target Task: Can GPT-3 see? | Conceptually Consistent |
|---|---|---|---|---|
| **Model 1** | correct | wrong | wrong | ✓ |
| **Model 2** | correct | correct | wrong | ✗ |
| **Model 3** | correct | correct | correct | ✓ |

Figure 1: Example target question with relevant and irrelevant background knowledge. A model is conceptually consistent when its knowledge of relevant background information – sharing concepts with the target – is consistent with its ability to answer questions correctly.

We want to build models for which that kind of understanding is possible, so in this work we take a step toward that goal by modeling the conceptual knowledge of a LLM and predicting when the LLM will be correct from that model. Our conceptual model is based on a LLM's answers to questions about background knowledge relevant to a particular anchor task (e.g., question answering), which we assume to be a reasonable though imperfect measurement of what the LLM can be said to "know." From this and a measurement of question answering performance we compute *conceptual consistency* (Figure 1), quantifying whether a model's knowledge of relevant background is consistent with its ability to perform the anchor task. Unlike standard approaches to evaluation this approach relies on example specific context.

Defining background knowledge is key to our approach because we need it to be relevant enough to establish meaningful consistency. Given the target query "Can GPT-3 see?" a relevant background query might be "Was GPT-3 built by OpenAI?" while "Is the sky blue?" would be an irrelevant query. Instead requiring a background fact to logically support the target query in some way, we say a background fact is relevant if it can tell us something about how a typical human language user would respond. Given a ground truth response $Y$ to the target with human response $\hat{Y}$ and respective responses $Y_K$ and $\hat{Y}_K$ to a potentially relevant background fact we define relevance using a conditional probability. If $P(Y = \hat{Y}|Y_K = \hat{Y}_K) \neq P(Y = \hat{Y})$ then the background fact is relevant because it is not independent of the target and gives us information about whether the speaker will be right or wrong. Knowing GPT-3 was built by OpenAI makes it more likely that someone will also know GPT-3 cannot see because it is true and involves relevant concepts. While knowing the color of the sky is laudable, there's no conceptual overlap with GPT-3's ability to see. In this paper, we focus on this kind of conceptual relevance.

After extracting background knowledge we use prompting to measure how a given LLM handles the background knowledge and how it performs at the anchor task. For this we study three varieties of generative language models across multiple scales up to 66 billion parameters and use a majority vote style prompting procedure to maximize the robustness of our approach to linguistic variations.

Our core contributions are

- We extract conceptually relevant background knowledge with respect to examples from an anchor task and map them onto background knowledge questions.
- We use a novel majority vote style zero-shot prompting procedure applied to generative language models to measure LLM performance.
- We measure conceptual consistency, focusing on generative language models and showing that consistency is low though it does increase with model size up to 66 billion parameters.
- We report conceptual patterns in model behavior that fall out of our analysis.

## 2 RELATED WORKS

Much work has been devoted to studying the limits of large language models beginning with BERT Devlin et al. (2019). Typical evaluation of language models measures performance on datasets of labeled examples across a range of tasks, such as those that constitute the amalgamated BIG Bench benchmark (Srivastava et al., 2022). In general they are explained as simply mimicking the data they were trained with. Low level critiques question the ability of LLMs to reason logically (Zhang et al., 2022a) or pass adapted psychological tests of language in humans (Ettinger,

2020). High level critiques question the ability of LLMs to understand anything at all (Bender & Koller, 2020), though one alternative avoids this by defining meaning based on conceptual role (Piantadosi & Hill, 2022).

**Consistency** Most relevant here is the large literature that studies failures in consistency of these models. A common approach is to verify an expectation about how the outputs of two different prompts should be related. For example, the prompts "Is the rose red?" and "What color is the rose?" with the expectation that answering "no" to the first question is inconsistent with answering "red" to the second (Ribeiro et al., 2019). CheckList (Ribeiro et al., 2020) offers a taxonomy that can be applied to these approaches. Invariance based consistency considers two versions of the input (often paraphrases) where the output is expected to be the same for both versions (Kassner & Schütze, 2020; Elazar et al., 2021; Ray et al., 2019; Jang et al., 2021; Ribeiro et al., 2020). Directional expectation consistency expects a specific change in the output based on how the inputs are constructed (Ribeiro et al., 2019; Wu et al., 2021; Kassner & Schütze, 2020; Jang et al., 2021; Ribeiro et al., 2020) – e.g., that one output should be the negation of the other. These works generally find that the models they study are often inconsistent, with varying degrees for various kinds of consistency. Many also improve their consistency measure by fine-tuning the studied models for it.

These kinds of consistencies are constructive rather than conceptual. They establish and improve the degree to which language models can be said to have beliefs (Hase et al., 2021), but in order for beliefs to support a theory of mind they should be actionable in that they should support predictions about another agent's behavior (Rabinowitz et al., 2018; Dennett, 1991). That a LLM has logically consistent outputs does not imply it will be right, but a theory of mind can provide evidence for or against correctness. This looser definition of consistency also makes the idea more general. In contrast, previous work, especially for more complex definitions of consistency, has relied on domain specific and expensive annotated datasets to measure consistency.

Furthermore, in this work we focus on large generative language models as opposed to the masked language models where most prior work on consistency is focused. Some of the greatest success has come from scaling generative LLMs (Hoffmann et al., 2022; Wei et al., 2022a), however most existing work on consistency focuses on smaller scale models (mostly masked style language models) because large scale generative models were not openly available until recently (Zhang et al., 2022b), and also because fine-tuning the larger models is prohibitively expensive, making it infeasible as a way to improve consistency. It may be that even though performance increases with scale consistency does not, so in this work we study model scale to the degree possible with open source models and a limited compute budget.

**Explanations and Interpretability** Explaining neural net decisions has been studied for a long time, but recent work has shifted focus towards concept based explanations (Yeh et al., 2021). For example, derivatives can be used to quantify conceptual influence in neural nets (Kim et al., 2018) and model the causal importance of concepts (Goyal et al., 2019). Another line of work links theory of mind to explanations (Chandrasekaran et al., 2017; Shvo et al., 2020). We use concepts to explain LLMs from a theory of mind perspective.

## 3 PROPOSED METHOD

To measure conceptual consistency we need to measure background knowledge and QA performance then predict the latter from the former. First we describe extraction of background knowledge in the form of questions with known answers, second we describe how we use prompting to answer both background and anchor questions, and finally we describe the conceptual consistency metric which correlates the two by predicting one from the other.

### 3.1 BACKGROUND KNOWLEDGE EXTRACTION

Here we focus on question answering (QA) problems and assume a knowledge base of content relevant to the specific QA task is available. Examples in our QA dataset consist of a question $Q$ with corresponding choices $S = \{s_1, \ldots, s_{|S|}\}$, one of which is the correct answer $A$. These choices can be single words or short phrases. We call $(Q, S, A)$ the anchor query because our first task is to find conceptually relevant background knowledge with respect to that information.

The background knowledge for a given anchor corresponds to a list of facts in a knowledge base. We assume each fact $F = (c^1, r, c^2)$ is represented by two concepts $c^1$ and $c^2$, and a relation $r$ between those concepts. Our task is to extract a set $B = \{f_1, \ldots, f_{|B|}\}$ of facts conceptually relevant to the anchor and then map those facts onto questions that can be asked to roughly determine whether the LLM knows the background knowledge.

**Extracting Background Facts**    We conceive of the knowledge base as a graph that connects concepts as nodes and relations as edges. To extract a list of concepts from a given text snippet we employ basic tokenization then remove stop words and keep only nouns, verbs, adjectives, and adverbs, following the pipeline from Ma et al. (2019). We construct the set of concepts $C$ as all concepts that appear in any part of the anchor query $(Q, S, A)$ (including incorrect answers) and have overlap ($> 50\%$ of words match) with a concept word or phrase from the knowledge base. For two different concepts $c^1, c^2 \in C$ we consider all tuples from all paths length $L$ or less which connect those concepts in the knowledge base, forming a cloud of relational information which constitutes the background knowledge for the anchor given by the selected knowledge base.

In practice this list of background knowledge tuples is too large, so we need to restrict it to a more manageable yet still relevant list. We do this by setting the maximum path length $L$ to 1, essentially looking for concepts which appear in the anchor and are directly related to each other in the knowledge base. This case follows Ma et al. (2019), where they were interested in extracting knowledge tuples to be inserted into neuro-symbolic models.

**Background Questions**    In order to measure how well a LLM already knows the content captured by these tuples we automatically translate them into natural language questions. Consider a fact tuple $(c^1, r, c^2)$. We substitute its three variables into a natural language template designed for the specific relation $r$. For example, the tuple (`book`, `used for`, `school`) would be translated into "Are book used for school?" Note that because the tuple exists in the knowledge base we know the correct answer to be some version of "yes". These templates are included in Table 2 of the appendix.

**Relevance**    This construction is likely to result in facts that are more relevant than irrelevant because each background query will share at least one concept with the target. Consider a human's response $\hat{Y}$ to a target query given their response $\hat{Y}_K$ to a background query extracted by this procedure. By construction our background query shares at least one concept with the target query. Since we assume that answers to questions are reflective of knowledge, answering the background query correctly indicates some knowledge of the background concepts, so it also indicates some knowledge of at least one of the concepts in the target query. As a result we expect the difference $|P(Y = \hat{Y}|Y_K = \hat{Y}_K) - P(Y = \hat{Y})|$ for a human language user to be positive for most background queries. We do not claim that all background queries extracted will be highly relevant or that all background queries put together will always be enough to determine the LLM's response confidently. Nevertheless, we do think our background knowledge should be predictive of target responses. Future work could try to increase the relevance of extracted background knowledge.

**Negative Background Knowledge**    In all tuples so far the relation $r$ really does exist between $c^1$ and $c^2$, so the correct answer is always "yes." Language models are often biased towards certain outputs and in this case we found a "yes" bias to be particularly strong in some models, especially the smaller versions of OPT (Figure 6a). As a result those smaller models can outperform the larger ones even when they understand the background knowledge less, just because they are prone to saying "yes" to everything. We fix this by extracting negative background knowledge tuples – to which the correct answer is some version of "no" – to mirror the positive ones.

We frame the problem in a pairwise fashion: given a positive tuple $(c^1, r, c^2)$ the negative tuple generation task is to find an alternative $\bar{c}^2$ for which the correct answer is "no." The pairwise nature ensures that we measure background knowledge in a balanced fashion to best address the "yes" (or "no") bias issue. As depicted in Figure 2, we form a set of choices $\bar{C}^2$ that includes every concept $\bar{c}$ in the knowledge base that meets 3 criteria:

1. $\bar{c}$ does not form a valid tuple $(c^1, r, \bar{c})$,
2. $\bar{c}$ is not cyclic (i.e., not equal to to $c^1$), and
3. $\bar{c}$ is in the English dictionary.

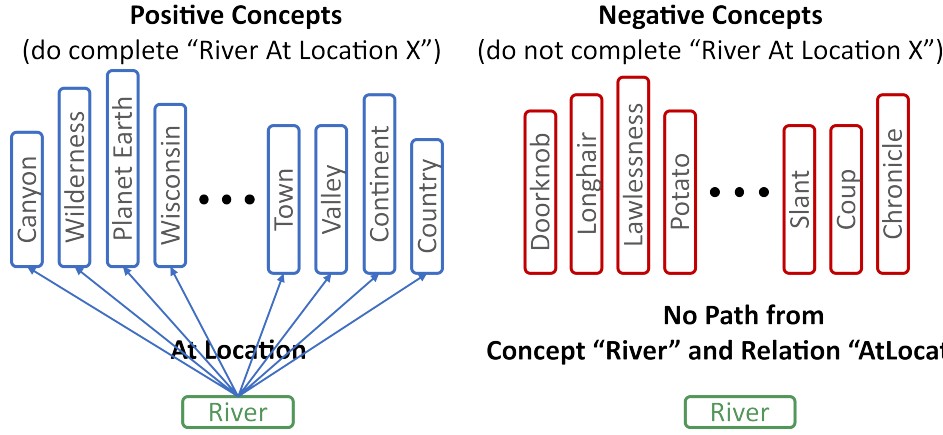

Figure 2: Facts from knowledge bases are true by virtue of being in the knowledge base. We mine negative background facts by finding concept pairs without an edge to ensure our extracted background knowledge is balanced.

Our final choice for $\bar{c}^2$ is a single sample from the uniform distribution over $\bar{C}^2$. The amount of concepts in the English dictionary is quite large, hence we curate the distribution space of $\bar{C}^2$ after applying criteria 1 and 2 to keep most frequently used concepts. We make this choice in advance, treating background tuples as a dataset, so that even if a positive background tuple is used for multiple anchor queries it is always paired with the same negative tuple. However, if there are multiple choices of $c^2$ for the same $c^1$ and $r$ then we sample $\bar{c}^2$ independently for each of those choices. The final set of background tuples for an anchor query includes all positive tuples that match that anchor along with all negative tuples that are paired with the positives. Examples of extracted negative background facts can be found in the appendix at Table 3.

## 3.2 ANSWERING BACKGROUND AND ANCHOR QUESTIONS WITH PROMPTING

Now we need to actually extract an answer to a given background question or anchor question from the model. We could simply provide the resulting text as input to the language model, see what most likely word it generates next, and take the result as the LM's answer. However, in experiments mainly with the OPT models we found this process to be highly noisy, with inconsistent performance across models and relations. This is probably due to inconsistencies in the models themselves (Elazar et al., 2021), but also the simple nature of our templating procedure which doesn't always get the grammar correct. We are interested in measuring the language model's *conceptual* consistency, not *linguistic* consistency, so we implemented a majority-vote-style procedure to make our prompting procedure more robust.

Instead of using just one question and one answer we consider many variations on the same question and many potential answers, taking the combination assigned the highest likelihood by the model as its predicted answer to the question. To vary the question presentation we substitute the question generated from our templates into 6 *meta-prompts*, which are minor variations on how to ask the question as reported in Table 1. To vary answer presentation we chose a list of 14 positive and negative words as potential answers including {(Yes, No), (True, False), (Right, Wrong), (Correct, Incorrect), (Positive, Negative), (Pass, Fail), (On, Off)}. This results in $6 \times 14 = 84$ model inputs for each query, and we evaluate the model's likelihood of each potential answer word. Note that no sampling is required because we only use single answer words. The final answer to a question is positive if the input with the highest likelihood used a positive answer word, and otherwise the answer is negative.

We found this variation to be essential for achieving some consistency across linguistic variations when experimenting with the OPT language models (Zhang et al., 2022b). Voting over multiple variations is similar to the strategies adopted in a number of recent approaches. In Shwartz et al. (2020) clarifying statements and answers to questions are chosen by taking a maximum over likelihood scores of different possibilities, but they do not use this to account for linguistic variation in prompts. Similarly, in Elazar et al. (2021) knowledge base tuples are completed by picking a choice from a restricted candidate set; here it is not used to control linguistic variation in order to study another variable, but is used to study linguistic variation itself. More recently majority vote has been used to evaluate generative models when generating longer snippets of text (Wang et al., 2022; Wei

Table 1: Meta-prompts utilized for question generation.

| Meta-Prompts |
|---|
| 1. `<question>`? |
| 2. `<question>`. Is this true? |
| 3. Answer this question as '`<label_a>`' or '`<label_b>`'. Question: `<question>`? |
| 4. Each item is a question and answer. Answer is one of '`<label_a>`' or '`<label_b>`'. Question: `<question>`? Answer: |
| 5. Pick '`<label_a>`' or '`<label_b>`'. Question: `<question>`?Answer: |
| 6. Question: `<question>`? Answer: |

et al., 2022b). These works use the approach to account for linguistic variation, but in the model outputs (more than just a few words) rather than in the inputs (i.e., the prompts). In comparison to all those studies, our majority-vote-style procedure is used to account for linguistic variation in the prompts themselves.

## 3.3 Measuring Conceptual Consistency

Now we are almost ready to measure conceptual consistency. Till now we have extracted answers $\hat{A}_B^{i,b}$ for the $b$th background knowledge question $Q_B^{i,b}$ for the $i$th example in the anchor task dataset. The anchor questions and answers are $Q_A^i$ and $\hat{A}_A^i$. We translate these questions and answers into scores that measure how well the model knows the background knowledge and how well it performs at the anchor task. These background and task scores are defined respectively *for each anchor example* using accuracy

$$S_B^i = \left( \mathbb{E}_{b \in \mathcal{P}_i} \left[ [[A_B^{i,b} == \hat{A}_B^{i,b}]] \right] + \mathbb{E}_{b \in \mathcal{N}_i} \left[ [[A_B^{i,b} == \hat{A}_B^{i,b}]] \right] \right) / 2 \tag{1}$$

$$S_A^i = [[A_A^i == \hat{A}_A^i]] \tag{2}$$

where $A_B$ and $A_A$ are the correct answers, $\mathcal{N}_i$ is the set of negative background tuples for anchor example $i$, $\mathcal{P}_i$ is the set of positive background tuples for anchor example $i$, and $[[.]]$ is the indicator function. Note that the background score weights negative and positive tuples evenly.

Finally we compute the conceptual consistency of a model on a given dataset and knowledge base by predicting the task score from the background score and reporting average precision

$$CC = AP(S_A, S_B) \tag{3}$$

where $AP(\cdot)$ measures the average precision of the $S_B$ predictor. Intuitively, this score will be high when the model answered more background knowledge questions correctly, so we are predicting that the model will perform better when it knows relevant background knowledge.

## 4 Experiment Setup

**Dataset and Knowledge Base** We conduct zero-shot evaluation on the CommonsenseQA (CSQA) (Talmor et al., 2019) task. CommonsenseQA covers a broad range of common sense concepts, with each entry containing a question and 5 answer choices. We choose ConceptNet as our knowledge base because CommonsenseQA is derived from it, although other knowledge bases would also be valid design choices. We use a subset of 14 out of the 36 available relations most relevant to common sense, including "antonym", "at location", "capable of", "causes", "desires", "form of", "has a", "is a", "made of", "part of", "related to", "similar to", "synonym", and "used for". Our experiments used the development set, since test set answers are not publicly available. No model training or hyperparameter tuning was applied to our pre-trained LLMs. Most of our development effort went into creating stable prompts.

**Models** We probe for conceptual consistency using three publicly available model families, including OPT-$\{350M, 1.3B, 13B, 30B, 66B\}$ (Zhang et al., 2022b), GPT(EleutherAI)-$\{125M, 2.7B, 6B\}$ (Black et al., 2022) and T0-$\{3B, 11B\}$ (Sanh et al., 2022). The OPT and GPT(EleutherAI) models were chosen because they are some of the largest publicly available models and have a range of checkpoints over sizes. We use T0 because it was tuned specifically for zero-shot prompting setting and achieved competitive results in zero-shot generalization. The prompts and general approach remain the same for all models, though we did most of the development using OPT models.

# 5 RESULTS

Here we report the conceptual consistency of the models we study, analyze the individual background and task components of that score, show qualitatively how performance on relations and concepts varies, and analyze bias related to our prompting approach.

**Conceptual Consistency** We compute the conceptual consistency (Equation 3) for each LLM and report the results in Figure 3. In absolute terms average precision is on the lower end, showing significant conceptual inconsistency. A model's knowledge of background information is somewhat predictive of its question answering correctness. Our results show that conceptual consistency generally grows with the scale of the model across all model classes,

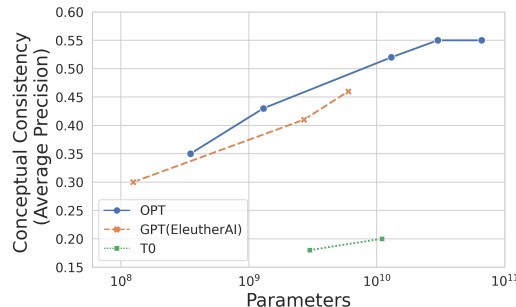

indicating that bigger models are not just more accurate, but also more consistent. A notable exception is between OPT-30B and OPT-66B, where there is no improvement from increased size. Both achieve the highest consistency among all models, but this one data point suggests there may be diminishing returns. OPT models perform best, with GPT models close behind and T0 demonstrating much worse performance. This indicates that OPT and GPT are much more conceptually consistent than T0. The difference could be due to the decoder only nature of OPT and GPT. It could also reflect a degree of forgetting in T0 due to supervised fine-tuning, but this is doubtful because one of

Figure 3: Conceptual Consistency of Language Models. This measures our ability to predict whether a language model will be correct from its knowledge of relevant background information.

the many datasets T0 was trained on was CSQA. However, it may also reflect the fact that our prompts are not designed specifically for T0.

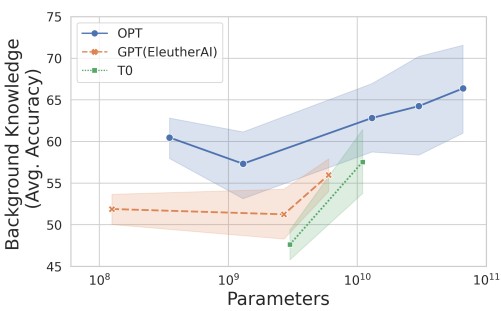
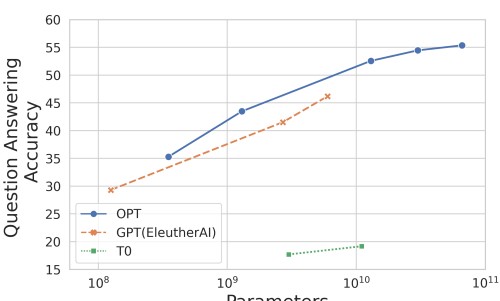

(a) Background knowledge performance: how good language models are at verifying whether our extracted background facts are true/false. Averaged over 14 relations.

(b) Anchor Task (CommonsenseQA question answering) performance measured by our zero-shot prompting approach.

Figure 4: Aggregated background performance and anchor task performance.

**Background and Task Performance** We also measure the components of conceptual consistency in Figure 4. For background knowledge we compute Equation 1 averaged per relation and then averaged over all 14 relations. This is reported with a 95% confidence interval in Figure 4a. There is an imbalance in how often these relations occur in the data, so this approach weights relations equally just for the purpose of assesing background knowledge. For task knowledge we report the anchor score (Equation 2) in Figure 4b, which is CSQA accuracy in these experiments. In both cases our prompting approach is able to extract correct answers from our LLMs. The trend in both cases is again an increase in performance with model size. This was expected in both cases, but it is interesting to note that the range of performance is smaller for background knowledge, suggesting that increasing size helps task knowledge more than background knowledge, though the opposite is

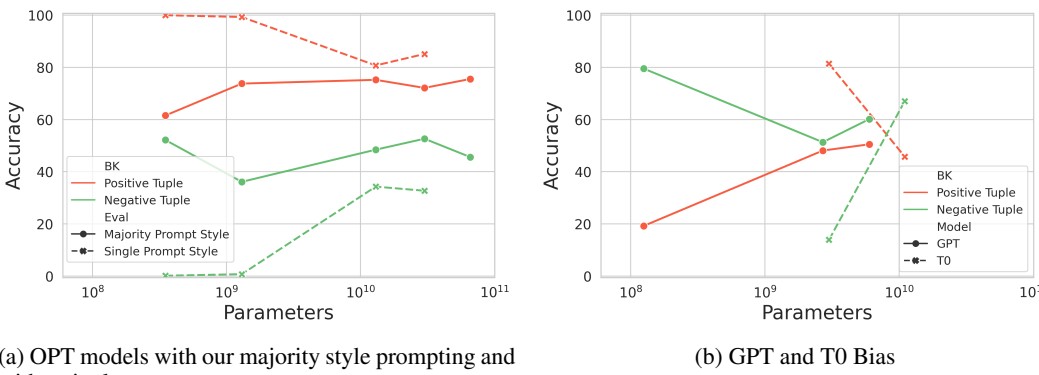

(a) OPT models with our majority style prompting and with a single prompt.

(b) GPT and T0 Bias

Figure 6: Background knowledge performance for negative and positive facts.

true for T0. Intuitively question answering is a more complex skill that builds on the less complex background knowledge completion skill. From that perspective these results are also an evidence of a skill hierarchy like Bloom's Taxonomy (Sahu et al., 2021), where simpler skills are learned before more complex ones. As for conceptual consistency, OPT models have higher performance compared to GPT and T0 models, and this is likely due to the same reasons discussed previously. It is however notable that the gap between T0 and the other models is much smaller for background knowledge. Also, we see a marginal increase in performance between OPT-30B and OPT-66B for both task and background knowledge, though it may not be significant.

**Background vs Consistency**
Next we ask where background knowledge and consistency diverge. In Figure 5 we plot background score versus conceptual consistency for 6 relations that seemed to be more like outliers. Smaller models tend to be further from the diagonal, either being inconsistent but knowledgeable or consistent without knowing much. Relations also behave differently. Small models don't know the "desires" relation very well, but they are somewhat conceptually consistent about it, to the point that even though large models get to

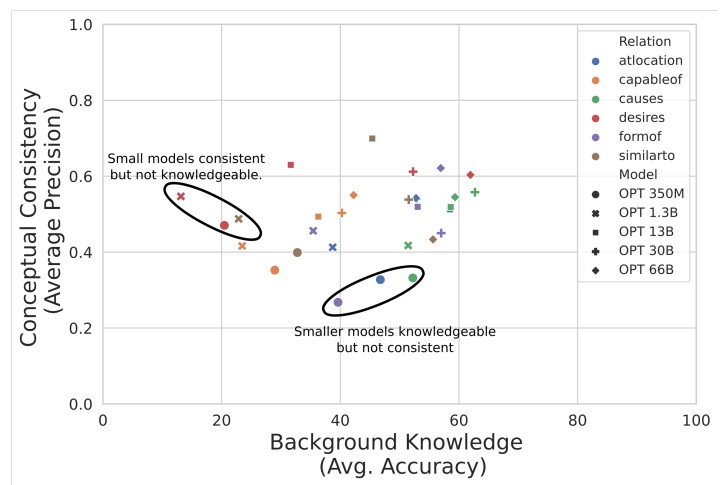

Figure 5: Scatter plot of Background Knowledge vs Conceptual Consistency.

know the relation better they do not get more consistent about it. In the reverse direction, all model scales know roughly the same amount about "causes" background information, but the larger models are much more conceptually consistent about it.

**Prompting Bias** In Figure 6 we analyze our majority-vote-style prompting and prompting biases we found in our LLMs. Each figure reports the background score (Equation 1) restricted to either the set of negative background facts or positive background facts, so accuracy reduces to the percentage of times the model said a version of "no" or "yes," respectively. Figure 6a reports this metric for the majority-vote-style prompting we used (Majority Prompt Style) and a previous version where we tried to find a single best meta-prompt and answer pair (Single Prompt Style). In our single prompt experiments we found that the two smaller OPT models had a strong "yes" bias as shown by the red dashed line in Figure 6a. We introduced negative background knowledge, which detected this problem (green dashed line), but we also found that our majority-vote-style prompting helped ameliorate the issue (solid lines). Even with majority-vote-style prompting both T0 and GPT(EleutherAI) still display a significant "yes" (T0) and "no" (GPT) bias at smaller scales.

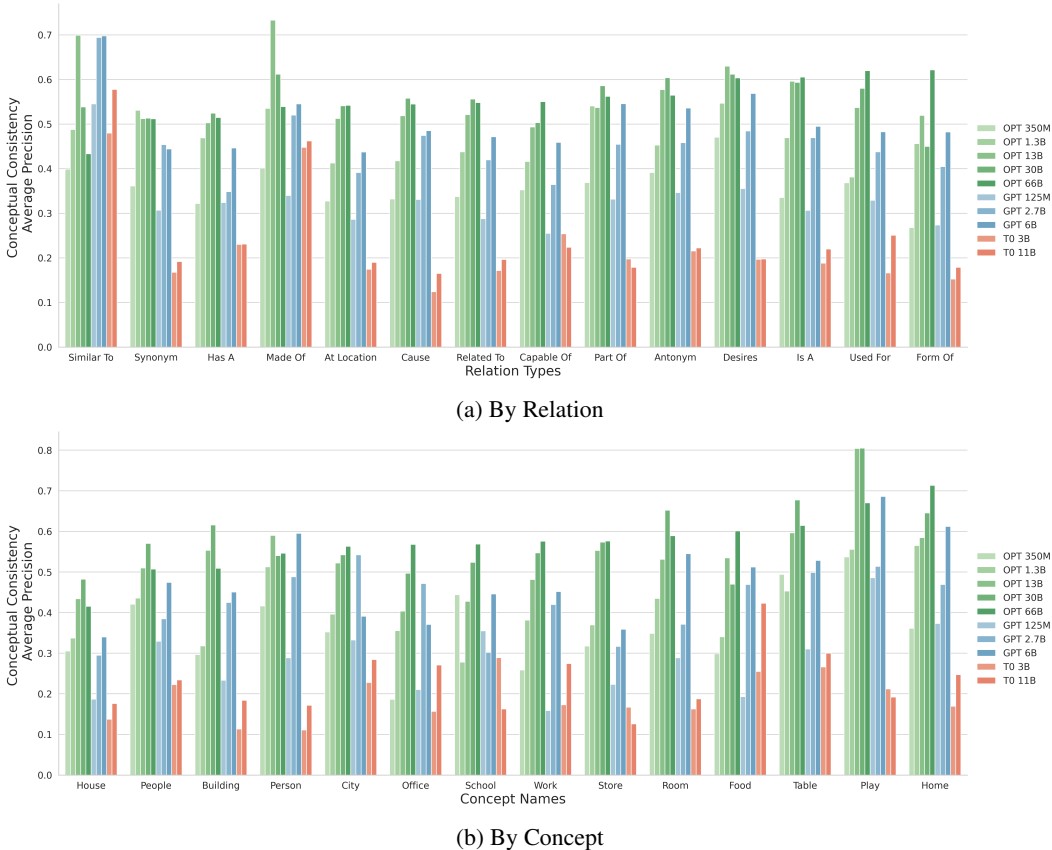

(a) By Relation

(b) By Concept

Figure 7: Conceptual consistency computed for different subsets of CSQA examples partitioned according to whether the background facts for an example contain the relation or concept.

**Concept and Relation Level Consistency**   We analyze the conceptual consistency at the level of relations and concepts in Figure 7. Figure 7a shows consistency for all relations. In case of Figure 7b, we picked 14 most occurring concepts with minimum occurrence count of 28. Though the overall trend is that larger models know more background knowledge, there are variations in which concepts and relations a model is consistent about. In some cases you don't need a larger model, or at least larger models can have sub-par performance. For example, our largest models (OPT-66B and OPT-30B) are outperformed by smaller versions of OPT, GPT(EleutherAI), and even T0 on the "Similar To" relation. Less extreme differences occur for other relations like "Made Of", and again lesser differences occur for concepts like "Play". This sensitivity of relations indicates our prompts, which we design per relation, could be a factor. Another observation is that the increase in performance with size is robust for GPT and T0, but that trend is more often violated for the OPT models. In general, trends in concept consistency are more robust than trends in relation consistency.

# 6   CONCLUSIONS

In this paper we built a kind of theory of mind about LLMs to study their conceptual consistency, whether their knowledge of relevant background information is consistent with their ability to answer questions correctly. For a question we extracted background knowledge from a knowledge base of related concepts and used prompting to measure whether popular open source LLMs knew that background information. We also measured their ability to answer common sense questions correctly. This set us up to measure conceptual consistency by predicting correctness from our background knowledge measure. We found that LLMs have a moderate amount of conceptual consistency, and that it increases with scale. We also found that while knowledge of background information increases with model scale it does not increase nearly as much as correctness or conceptual consistency, indicating that models size has a larger impact on difficult tasks than simpler ones and providing evidence of a hierarchy of related skills. In the future we would like to study whether this measure of consistency can be used to guide how humans understand LLMs. We also want to focus further on the hierarchy of skills by creating a dataset that tests multiple levels of comprehension.

AUTHOR CONTRIBUTIONS

ACKNOWLEDGMENTS

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

# A APPENDIX

We prepared different prompt templates for the picked 14 relations used in our experiments shown in Table 2.

Table 2: All the relations with its corresponding prompt style with a sample example.

| Relation | Prompt Style | Sample Instance |
|---|---|---|
| Is A | Is $c^1$ a $c^2$? | Is security a department? |
| Has A | Does $c^1$ has a $c^2$? | does house has a basement? |
| Antonym | Is $c^1$ an antonym of $c^2$? | Is clever an antonym of dull? |
| Cause | Does $c^1$ cause $c^2$? | does fencing cause small cuts? |
| Desires | Does a $c^1$ desires $c^2$? | does a dog desires affection? |
| Form Of | Is $c^1$ a form of $c^2$? | Is partying a form of of party? |
| Made Of | Is the $c^1$ made of $c^2$? | Is the car made of metal? |
| Part Of | Is $c^1$ a part of $c^2$? | Is book a part of library? |
| Related To | Is $c^1$ related to $c^2$? | Is doctor related to illness? |
| Similar To | Is $c^1$ similar to $c^2$? | Is ridiculous similar to silly? |
| Synonym | Is $c^1$ a synonym of $c^2$? | Is reply a synonym of answer? |
| Used For | Are $c^1$ used for $c^2$? | Are clothes used for wearing? |
| At Location | Is $c^1$ at location $c^2$? | Is door at location library? |
| Capable Of | Is a $c^1$ capable of $c^2$? | Is a child capable of form opinions? |

In Table 3, we showcase few instances of the negative background facts created from the process described in subsection 3.1.

Table 3: Examples of negative background knowledge task for each relation.

| Relation | Negative Background Facts |
|---|---|
| Is A | [Is drill a clamp?, Is ocean a shame?, Is space a micrometer?] |
| Has A | [does mammals has a watch?, does pen has a unicycle?, does oceans has a uncle?] |
| Antonym | [Is wash an antonym of detached?, Is bad an antonym of nightdress?, Is shade an antonym of improvement?] |
| Cause | [does going into coma cause company?, does standing in queue cause shrinking?, does doing housework cause mermaid?] |
| Desires | [does a person desires schizophrenia?, does a children desires unemployed?, does a tree desires criticism?] |
| Form Of | [Is recycled a form of burned?, Is bath room a form of interrupted?, Is storing a form of bleeding?] |
| Made Of | [Is the ocean made of uncomfortableness?, Is the car made of rain?, Is the plants made of bicycle?] |
| Part Of | [Is gulf a part of round?, Is shower a part of anger?, Is bell a part of congress?] |
| Related To | [Is class related to cornstarch?, Is room related to jordan?, Is cable related to tilemaking?] |
| Similar To | [Is lie similar to botany?, Is ridiculous similar to aspirin?, Is distant similar to physiology?] |
| Synonym | [Is heart a synonym of volition?, Is station a synonym of subordination?, Is part a synonym of undermine?] |
| Used For | [Are hair used for council?, Are theatre used for desk?, Are disk used for pain?] |
| At Location | [Is monkey at location fuzzball?, Is piano at location macaroni?, Is table at location gauging?] |
| Capable Of | [Is a computer capable of pillow?, Is a band capable of overmodulation?, Is a tiger capable of fireman?] |

