# OpenReview forum: "Unpacking Large Language Models with Conceptual Consistency"
_ICLR.cc/2023/Conference — Submitted to ICLR 2023_

### Official Review · Reviewer_nKxz · 2022-10-22

**Confidence:** 4
**Correctness:** 4
**Technical Novelty And Significance:** 3
**Empirical Novelty And Significance:** 3
**Recommendation:** 8

**Clarity, Quality, Novelty And Reproducibility:**

The paper is well written and almost error free, it addresses a current problem and is fit very well into the literature. While many works have a focus on constructive consistency, the present study, with a looser definition of consistency, aims at a more general relationship between background knowledge and "understanding" of language models. The proposed  measurement of conceptual consistency can be used on a wide variery of different language models, downstream tasks and datasets. The results should be easily reproducible, since the datasets and models used are public and the large language models have not been further trained or finetuned.

**Strength And Weaknesses:**

The basic idea of measuring consistency in a language model is well founded and comprehensible. The argumentation of the authors and the structure of the experiments can by and large be followed very well. It becomes clear that specific conceptual consistency in large generative language models has not yet been comprehensively considered, and findings in this area could provide clues as to how models might arrive at a better understanding of language. The results of the measurements carried out are convincingly presented and discussed. The paper is overall well written with few negligible typos.

Although the extraction of background knowledge from the language model and the generation of the background questions is described in great detail, it might be possible to describe more clearly how the concepts to the anchor questions of the QA dataset are selected: The set of concepts is "all the [...]words and phrases that appear in any part
of the anchor query". Since the anchor question (Q,S,A) also includes the incorrect answers, I assume that only concepts from Q and A (not S) are considered? If the mapping of a phrase to the concept of the knowledge base is done by a 50% words match, could this possibly lead to errors?
Since the results of the conceptual consistency and the accuracy of the QA task are very similar (Fig. 3/Fig. 4(b)), although different things are measured, it might be interesting to show the consistency on the basis of one a different task also.

minor comments:
•	order of figures 6, 5 (p. 8)
•	typo punctuation "perform better at this Now conceptual" (p. 6)
•	cite parentheses "BERT Devlin et al." (p. 2)
•	references: all authors shown (p. 12!)


**Summary Of The Paper:**

Although current large-scale language models are already being used very successfully to complete many NLP tasks, sometimes with surprisingly good results, they are criticised for not understanding language but only imitating the data on which they have been trained. In order to understand where the limitations of these language models lie and how they can perhaps be overcome, it is all the more important to be able to interpret and explain the decisions of these neural networks against the background of what knowledge they can actually draw on. The authors therefore propose a model to check whether the relevant background knowledge of a language model corresponds to its ability to perform a particular task. To do this, they design a measure of conceptual consistency that relates background conceptual knowledge extracted from the language model to performance on a QA task. The paper examines conceptual consistency for language models of different sizes, finding  that although background knowledge increases with model size, it does not increase as much as consistency or performance on the task. The authors therefore assume that correctly answering questions about the background knowledge of a language model requires less skill than the QA task and would be at a lower level in a hierarchy.

**Summary Of The Review:**

The authors propose a measure for examining language models that they can justify well and show its value in a sample study. Interesting (although perhaps not particularly surprising) results are also obtained by looking at the individual components, the background knowledge of the model in comparison with its performance on the downstream task. The results of this and further studies could provide clues to a taxonomy of skills in language models.

---

> ### Author Response · Authors · 2022-11-18
> **Response to nKxz**
>
> > it might be possible to describe more clearly how the concepts to the anchor questions of the QA dataset are selected
>
> Indeed we could have been more clear here and in other sections of the approach. We have improved this clarity in the latest revision. In the Extracting Background Facts portion of section 3.1 we address this specific concern: “To extract a list of concepts from a given text snippet we employ basic tokenization then remove stop words and keep only nouns, verbs, adjectives, and adverbs, following the pipeline from Ma et al. (2019).” We do include the incorrect answers when we extract concepts from the anchor query. They could still contain relevant concepts that might appropriately decrease the likelihood a model places on the incorrect answers, thus helping choose the correct one.

---

### Official Review · Reviewer_w16h · 2022-10-23

**Confidence:** 3
**Correctness:** 2
**Technical Novelty And Significance:** 2
**Empirical Novelty And Significance:** 3
**Recommendation:** 3

**Clarity, Quality, Novelty And Reproducibility:**

Apart from the weakness cited above, the paper's writing is sufficiently clear and easy to follow. The quality and reproducibility of the work is hard to judge due to a lack of shared (open source) code. As discussed above, this paper does have novelty in terms of the formulation of the conceptual consistency and the way in which the author probe what background knowledge an LLM possesses, although a lot is left to be desired in the detailed technical implementation of the ideas.

**Strength And Weaknesses:**

Strengths of the paper:
1. The formulation conceptual consistency of LLM as a binary prediction problem for answers to the anchor CSQA questions based on answers to background-knowledge questions.
2. The author made the observation that the LLMs show a strong bias towards certain answers to binary questions (preferring "Yes" to "No") and may prevent a reliable extraction of background knowledge. The authors invented the novel approach that combines 1) a method for creating negative examples from the ConceptNet graph, 2) using different wording in the prompt when probing background knowledge, and 3) relying on a majority voting approach. The authors quantified the improvement of background-knowledge probing of this novel approach compared with a single-prompt baseline.

Weaknesses
1. While the formulation of the conceptual consistency is practical and clear, it only covers a narrow aspect of self-consistency that determines whether "theory of mind" (in the author's wording) can be applied to LLMs. Due to the operational definition of the conceptual consistency, the method only captures the model's knowledge with respect to the existence of a relatively small number of (14) relations. In addition, the answers of the LLMs capture only the binary existence of a relation, with no attention paid to other information, such as whether a certain relation's existence is not clearly binary and may dependent on other factors.
2. The way in which the authors selected background-knowledge questions from the ConceptNet is not justified clearly enough. In particular, the authors used a maximum path length of 1 in the knowledge graph, which essentially limits the background-knowledge questions to concepts that are present only in the anchor question itself. However, it is conceivable and likely that answering a certain anchor question depends on knowledge beyond the concepts that are present in the anchor questions. Take the example that author gave in the introduction, the fact that "The peak of a mountain almost always reaches above the tree line" relates to concepts such as "tundra", "tree line", and "height" that are not present in plain text form in the question's text. Therefore the authors should have included a more careful analysis of how varying the maximum path length in the ConceptNet affected the consistency scores. It should be possible to at least experiment with small values of maximum path length to stay within practical computational budget.
3. The hand-engineered meta-prompts used to probe background knowledge is not sufficiently justified. It has been shown previously that model fine-tuning and soft prompt tuning (https://arxiv.org/abs/2104.08691) out perform hand engineering of text prompt. It is likely that the LLMs can be more reliably probed for the background knowledge by fine-tuning or prompt-tuning based on a small number of (O(100)) unambiguous and basic examples. Alternatively, few-shot prompting can also be explored.
4. The paper lacks some important technical details. The most important ones are how the sampling of the LLMs were performed, e.g., what temperature was used and whether greedy sampling or beam search was involved.
5. A remarks in the Results section are not well supported.  Specifically, the authors claim that the LLMs show a wider range in the their background knowledge than their question-answering accuracy (Figure 4). This claimed isn't supported by quantitative or statistical analysis and hence seems anecdotal. It also doesn't seem to be true even based on the average curves alone in Figure 4.

**Summary Of The Paper:**

This paper attempts to measure the conceptual consistency of the current large language models (LLMs) in question answering tasks of the sort exemplified in the CommonsenseQA (CSQA) dataset. Specifically, the authors focused on the consistency between an LLM's answers to binary (yes/no) questions regarding the existence of relations between relevant concepts and the the answer to multiple-choice questions from the CSQA dataset. The authors claim the finding that the current open LLMs including OPT, GPT-3 (EleutherAI), and T0 have low conceptual consistency (in the range of 0.15 to 0.55 average precision), but the consistency score increases with model size.

**Summary Of The Review:**

In summary, this paper attempts to address an important and interesting question of how self-consistent LLMs are in their answers to knowledge-based commonsense questions. It contains the novel ideas of formulating the conceptual consistency problem as a binary prediction problem based on probing relevant background knowledge through binary questions synthesized from ConceptNet. It also proposes an interesting approach to synthesize negative questions and using majority voting to gauge the model's background knowledge more reliably. However, the paper did not carefully implement the knowledge measurement, leaving out a thorough exploration of path length on ConceptNet and alternatives ways of eliciting model outputs.

---

> ### Author Response · Authors · 2022-11-18
> **Response to w16h**
>
> > While the formulation of the conceptual consistency is practical and clear, it only covers a narrow aspect of self-consistency…
>
> We agree that our experiments implement a somewhat narrow version of conceptual consistency because of the limited relations (14) and question types (binary only). Our approach does not try to capture all or even most relevant background knowledge. Conceptual consistency is meaningful as long as the background knowledge has some relevance, because in a human it should still be predictive of model correctness.
>
> Furthermore, these design choices were made to maximize the potential relevance of the background we measure in our chosen domain. Limited relations and binary questions are not a general limit on the conceptual consistency approach, but design choices we made to simplify the implementation.
>
> > The way in which the authors selected background-knowledge questions from the ConceptNet is not justified clearly enough.
>
> We have discussed this concern at length and improved our justification of the background extraction procedure in our recent revision. Our idea is NOT that all background knowledge will be relevant or that all relevant background will be captured. This is described in our overall response to reviewers and in our response to reviewer dh2f. Briefly, we define our notion of relevance using the probability of correctness on a target query given correctness on a relevant background query. We then argue that this notion of relevance is typically satisfied for the background facts we extract because they share concepts (certain words in common) with the target query.
>
> > The paper lacks some important technical details. The most important ones are how the sampling of the LLMs were performed, e.g., what temperature was used and whether greedy sampling or beam search was involved.
>
> Because answers to questions are single words, we do not sample our LLM’s at all. Instead we just compare the logits of the potential answer words. We have clarified this in section 3.2.
>
> > hand-engineered meta-prompts used to probe background knowledge is not sufficiently justified [vs fine-tuning or prompt tuning]
>
> Our approach aims to just evaluate these models and not do any training specific to the domain of choice. Meta-prompts allow us to do that, but fine-tuning and prompt tuning do not. Introducing the proposed methods would require introducing training data for both the background knowledge and the proposed task, while hand engineered meta-prompts prevent divergence between evaluation procedures. However, we agree that using these models in a few-shot (rather instance based learning) setting could be an interesting alternative design choice.
>
> > A remarks in the Results section are not well supported. Specifically, the authors claim that the LLMs show a wider range in the their background knowledge than their question-answering accuracy (Figure 4).
>
> For OPT, background Knowledge (Figure 4.a) ranges from about 57% to 66% (a difference of 9), whereas QA accuracy (Figure 4.b) ranges from 35% to 55% (a difference of 20). We were referring to these differences (9 vs 20) in that remark, so we think this statement is quantitatively supported. For GPT(EleutherAI) these differences are approximately 4 (background) and 17 (QA) and for T0 they are 10 (background) and 2 (QA), so the same observation holds for GPT, but not T0. We’ve updated the paper to point out the latter exception.

---

> > ### Comment · Reviewer_w16h · 2022-11-28
> > **Thanks to the author for addressing my comments**
> >
> > I thank the authors for revising the manuscript according to my comments and comments from the other reviewers, and for replying to my points above. The revised manuscript is generally improved from the original version due to the added information and discussion, which helps to clarify some of the doubts from myself and the other reviews.
> >
> > Some of my questions have been addressed, but two questions remain.
> >
> > First, in Section 3.1, the authors claim "In practice this list of background knowledge tuples is too large, so we need to restrict it to a more
> > manageable yet still relevant list", hence the authors choose to stay with "path length equal to 1". This claim is not supported by  quantitative data. This is related to my previous comment "Therefore the authors should have included a more careful analysis of how varying the maximum path length in the ConceptNet affected the consistency scores." In general, I feel the authors should provide more quantitative details in order to satisfactorily address this comment. For example: what is the average number of background relations at path length = 1 (per anchor query)? What would the number be at path length = 2? By going from path length = 1 to path length = 2, how much increase in the computational cost will it entail (how many times x?). If the increase in computational cost is too dramatic at path length = 2, then the authors could take a middle-of-the-road approach by randomly sampling a set of the relations at the 2nd-degree relations. Has this been considered or at least discussed?
> >
> > This also brings up a question of how the 14 relations were chosen from the 36 total relations available in ConceptNet. Currently the manuscript doesn't have a justification of the choice, except "most relevant to common sense", which is unclear.
> >
> > Second, regarding the pros/cons of hand-engineered prompts, few-shot prompting, and fine-tuning/prompt-tuning in probing the background knowledge, there needs to be some discussion added to the manuscript.

---

### Official Review · Reviewer_yTJs · 2022-10-24

**Confidence:** 3
**Correctness:** 2
**Technical Novelty And Significance:** 3
**Empirical Novelty And Significance:** 3
**Recommendation:** 3

**Clarity, Quality, Novelty And Reproducibility:**

The construction of the cornerstone of the theory is really fuzzy. Section 3.1. seems to be a very important spot in this paper. However, it is not clear at all how background knowledge is created from the initial (Q,S,A) triple. This seems to be related to facts F=(c^1,r,c^2), but it is not clear how. Moreover, there are other symbols f in a set B, which is not clear what they are (pag.4)

**Strength And Weaknesses:**

Strength
- an interesting method to explore real capabilities of pre-trained language models

Weaknesses
- many details of the method are not clear
- it is difficult to asses the quality of results
- the paper is not well organized

**Summary Of The Paper:**

The paper describes a method to evaluate how LLMs learn about knowledge.
The model seems to be based on the following intuition. Given a task item, a set of background facts is extracted. If the LLM can solve the background facts, then it can solve the task item. This is a clever idea.

**Summary Of The Review:**

The paper hides important content. Yet, it should be written more clearly In order to let the reader reach this content

---

> ### Author Response · Authors · 2022-11-18
> **Response to yTJs**
>
> > it is not clear at all how background knowledge is created
>
> Indeed we could have been more clear about background knowledge. We have improved this clarity in the latest revision. In the Extracting Background Facts portion of section 3.1 we address this specific concern: “To extract a list of concepts from a given text snippet we employ basic tokenization then remove stop words and keep only nouns, verbs, adjectives, and adverbs, following the pipeline from Ma et al. (2019).” We do include the incorrect answers when we extract concepts from the anchor query. They could still contain relevant concepts that might appropriately decrease the likelihood a model places on the incorrect answers, thus helping choose the correct one.
>
> > many details of the method are not clear
>
> We have also updated the rest of the approach to improve clarity and made small changes throughout the rest of the paper including figure 1 and our toy example.

---

### Official Review · Reviewer_dh2f · 2022-10-29

**Confidence:** 4
**Correctness:** 3
**Technical Novelty And Significance:** 2
**Empirical Novelty And Significance:** 2
**Recommendation:** 3

**Clarity, Quality, Novelty And Reproducibility:**

The writing is mostly clear. In terms of technical quality, there are missing links between what the paper attempt to conclude on and the indicator/proxy measures which this paper investigates for LLMs.  The idea to investigate what LLMs knows or how to determine their trustfulness regarding given tasks is inspiring but not complete yet.


**Strength And Weaknesses:**

Strength
	• An attempt to evaluate LLMs  through a metaphor of conceptual consistency (it seems that the authors want to evaluate a form of reliability or trust by looking at what LLMs' "knows" but no rigorous definitions are given)
	• The definition of conceptual consistency itself is well-defined in the form


Weakness
	• It seems that the authors want to evaluate a form of reliability or trust by looking at what LLMs' "knows" but no rigorous definitions are given.
	• It is not clear that whether Conceptual consistency between LLMs' CSQA and the QA question-answers extracted from ConceptNet really reveals "what LLMs know". There are heuristic syntactic procedures to extract QA background facts from ConceptNet for CSQA question-answer pairs. The influence of heuristic procedures needs to be formally characterized and bounded for the "what LLMs know" conclusion to be rigorously valid.  There is a logical gap between "what LLMs know" and the "conceptual consistency definition" as it is now.


**Summary Of The Paper:**

This paper compares pretrained LLMs' question-answer results for CSQA dataset and ConceptNet. The study is on comparing the QA performance between CSQA dataset and QA problems extracted from ConceptNet through a relevancy heuristic procedure.   The authors attempt to interpret the average precision metrices comparison of the two related QA datasets as conceptual consistency.


**Summary Of The Review:**

This paper investigates into an important problem of LLMs what LLMs knows or how to determine their trustfulness regarding given tasks are inspiring but not complete yet.  This paper attempts to understand what LLMs know by looking at the conceptual consistency between QA datasets and the underlying knowledge graph. However,  it is not clear that whether Conceptual consistency between LLMs' CSQA and the QA question-answers extracted from ConceptNet really reveals "what LLMs know" thus reveals whether LLMs are trustworthy for given tasks.

Details
	• Page 4, please investigate formal means to evaluate whether the extracting background facts are truly background facts corresponding to the queries or answers. This might be a formal means of relevancy beyond syntactic or linguistic heuristics but more in terms of semantics.
	• Page 5, please quantify the influence of prompting engineering in relevance to the conceptual consistency performance and the variants of how the pretrained LLMs was obtained (e.g. training data)
	• Page 6, models: Please provide more training dataset details of the tested LLMs. In particular, the training datasets and distribution of topics (or meanings in general) might highly correlate with their performance over the problem sets created from the QA pairs or concept tuples from ConceptNet --- informally the coincidence and difference of the two distribution of meanings in LLMs' training data and ConceptNet.

---

> ### Author Response · Authors · 2022-11-18
> **Response to dh2f**
>
> > There is a logical gap between "what LLMs know" and the "conceptual consistency definition" as it is now.
>
> Your main concern was that our current definition of conceptual consistency does not really capture what a LLM "knows" and thus is not very useful as a form of reliability or trust and cannot support our conclusions. Instead, the review argues, the connection between CSQA performance and background facts is too heuristic and that the background extraction procedures “need to be characterized and bounded to make conclusions about what [the] LLM knows.”
>
> This is an important criticism which we discussed extensively and have addressed in our paper revision. Previously we characterized our background knowledge by contrasting constructive relevance and conceptual relevance, but have now revised the paper to give a more precise definition. Specifically, in the introduction we specify that background knowledge is relevant when the difference between the probability of correctness given background knowledge is different from a prior probability of correctness. In section 3.1 we then argue that our background extraction procedure results in relevant background facts because our extracted facts are constructed in such a way that they share concepts with the target query.
>
> > This paper investigates into an important problem of LLMs what LLMs knows or how to determine their trustfulness regarding given tasks are inspiring but not complete yet.
>
> Of course we cannot solve the problem of determining when to trust a LLM here, but the issue seems to be about whether this contribution is at a point where it will benefit the community, which goes back to the previous point about background knowledge extraction. We think our notion of consistency is indeed meaningful and novel, so it could both inspire future work and help readers understand LLMs better.
>
> > Please provide more training dataset details of the tested LLMs
>
> Our approach does not use any training or fine tuning, so there is no training dataset to report details of. We refer the reviewer to the individual papers for the models we use for details about their pre-training.

---

### Author Response · Authors · 2022-11-18
**Overall Response to Reviews**

We thank the reviewers for their thoughtful comments, time, and effort. Here we summarize and respond to the major points raised by the reviews as a whole. We have also responded to individual reviews below and uploaded a revised version of the paper to address these concerns.

The main concern from the reviews was about explaining and justifying our background knowledge extraction procedure. In our revision and responses to reviewers we have both clarified our presentation and expanded on our justification. Previously we characterized our background knowledge by contrasting constructive relevance and conceptual relevance, but have now revised the paper to give a more precise definition. Specifically, in the introduction we specify that background knowledge is relevant when the difference between the probability of correctness given background knowledge is different from a prior probability of correctness. In section 3.1 we then argue that our background extraction procedure results in relevant background facts because our extracted facts are constructed in such a way that they share concepts with the target query.

Another concern was about the clarity of the paper. Our edits have largely been aimed at improving the clarity of the paper and we address specific reviewers in our individual responses.

We would also like to highlight one about how our approach works. At no point do we perform any training, and this is by design to keep our approach general. If we were to engage in fine-tuning or prompt tuning then that would require us to collect training data for both background and target tasks and somehow balance the two during fine-tuning. It’s not clear what fine-tuning or prompt tuning would say about the knowledge of the original model. Furthermore, because we’ve constructed single answer questions we also do not sample from these LLMs; we only compare the logits of the possible answer choices.

---

### Decision · Program_Chairs · 2023-01-20

**Decision:**

Reject

**Justification For Why Not Higher Score:**

While the paper addresses a very relevant question, the reviewers express doubts about whether the used metric actually gives us insights. The experiments are also limited and, more importantly, it is not clear why this is the case, e.g. why not testing on all relations on ConceptNet.

**Justification For Why Not Lower Score:**

N/A

**Metareview: Summary, Strengths And Weaknesses:**

The paper describes a new metric for accessing how well language models can understand concepts. The authors use this metric to analyze the performance of language models on common sense reasoning tasks. A main finding is that most language models do not exhibit strong conceptual consistency.

Strengths:
- Relevant problem space
- Interesting formulation of conceptual consistency as a classification problem

Weaknesses:
- Doubts about whether the conceptual consistency metric used in the paper indeed gives more insights into what language models know
- Somewhat limited novelty of the method
- Coverage of only a relatively small number of relations (14) and no description of why not all (36) relations of ConceptNet were used
- Lack of clarity in the paper for background knowledge inclusion (improved through revision) and some technical details such as sampling strategies and tuning approaches